

# The accuracy-bias trade-offs in AI text detection tools and their impact on fairness in scholarly publication

Ahmad R. Pratama

University Libraries, State University of New York at Stony Brook, Stony Brook, New York, United States

## ABSTRACT

Artificial intelligence (AI) text detection tools are considered a means of preserving the integrity of scholarly publication by identifying whether a text is written by humans or generated by AI. This study evaluates three popular tools (GPTZero, ZeroGPT, and DetectGPT) through two experiments: first, distinguishing human-written abstracts from those generated by ChatGPT o1 and Gemini 2.0 Pro Experimental; second, evaluating AI-assisted abstracts where the original text has been enhanced by these large language models (LLMs) to improve readability. Results reveal notable trade-offs in accuracy and bias, disproportionately affecting non-native speakers and certain disciplines. This study highlights the limitations of detection-focused approaches and advocates a shift toward ethical, responsible, and transparent use of LLMs in scholarly publication.

## INTRODUCTION

In recent years, there has been an unprecedented rapid development in the field of artificial intelligence (AI), particularly with the surge in popularity of large language models (LLMs) across various domains. Shortly after OpenAI released ChatGPT in late 2022, several other LLMs became available to the general public, including Google's Bard, which was rebranded as Gemini one year after its launch. Within a brief period, LLMs have become the next big thing, and their use has become widespread and mainstream, disrupting numerous aspects across multiple domains. LLMs have also significantly impacted academic and scholarly publication. Researchers increasingly use LLMs for information seeking, editing, ideation, framing, direct writing, data cleaning, data analysis, and even data and content generation (*Liao et al., 2024*).

This widespread adoption, however, raises some ethical concerns. In some cases, authors have been caught directly copying and pasting content generated by LLMs into publications without disclosure (*Kendall & Teixeira da Silva, 2024*), some of which have led to retractions (*Lei et al., 2024*). Other cases involve the controversial listing of LLMs as co-authors in scholarly works (*Nazarovets & Teixeira da Silva, 2024*; *Stokel-Walker, 2023*; *Yeo-Teh & Tang, 2024*), which has sparked debates regarding accountability and authorship. Eventually, this led to a consensus that LLMs cannot be listed as authors of

Corresponding author
Ahmad R. Pratama,
ahmad.pratama@stonybrook.edu

scholarly articles, but their use as assistive tools to improve the readability of writing is acceptable (*Lund & Naheem, 2024*). Furthermore, some journals and publishers have modified their guidelines to address the use of generative AI and LLMs (*Cheng & Wu, 2024*; *Ganjavi et al., 2024*; *Flanagin et al., 2023*). However, not all researchers acknowledge LLM involvement in their scholarly articles (*Pesante, Mauffrey & Parry, 2024*). Some argue that it is unnecessary to disclose LLM involvement when it is solely used as a tool to improve readability, either by correcting spelling and grammar or other general editing (*Chemaya & Martin, 2024*), while others hesitate due to fear of stigma or misconceptions about their work's originality and integrity (*Giray, 2024a*). This further complicates the ethical landscape of LLM adoption in academia and scholarly publication.

Historically, non-native English-speaking scholars have faced significant challenges in navigating the scientific communication realm, where English is the international language of scholarly publication (*Amano et al., 2023*; *Cho, 2004*; *Ferguson, Pérez-Llantada & Plo, 2011*; *Flowerdew, 2001*; *Horn, 2017*; *Kojima & Popiel, 2022*; *Raitskaya & Tikhonova, 2020*). Nearly all reputable journals require that manuscripts conform to elevated linguistic standards, often necessitating professional proofreading services that can be prohibitively expensive (*Amano et al., 2023*; *Mumin, 2022*; *Ramírez-Castañeda, 2020*; *Van Noorden, 2013*). This is where LLMs can play a key role by providing a transformative and unprecedented opportunity to equalize access, assisting researchers in improving the language, style, and clarity of their work (*Liao et al., 2024*). For these researchers, whose first language is not English, LLMs can offer a cost-effective alternative to traditional editing services, leveling the playing field and promoting fairness in scholarly publication.

Meanwhile, it is well-known that different disciplines have distinct writing styles when it comes to scholarly publication, reflecting the varied conventions, methodologies, and epistemologies unique to each field (*Alluqmani & Shamir, 2018*; *Dong et al., 2024*; *Dong, Mao & Pei, 2023*). This diversity, however, raises the possibility of bias in AI-generated text produced by LLMs, which may perform better in disciplines with simpler, more standardized language structures while struggling with the nuanced and often interpretive styles found in disciplines like the humanities, social sciences, and interdisciplinary journals. Furthermore, concerns have been raised about the potential for LLMs to drive both language use and knowledge construction toward homogeneity and uniformity (*Kuteeva & Andersson, 2024*). By reproducing existing patterns in training data, LLMs risk amplifying pre-existing biases and structural inequalities embedded within scholarly publication. This could inadvertently reinforce dominant paradigms while marginalizing alternative voices and perspectives, undermining the diversity that is critical to advancing knowledge across disciplines.

On the other hand, AI text detection tools have emerged as a means to address concerns about undisclosed use of AI, especially in academic settings (*Perkins et al., 2024*; *Mitchell et al., 2023*). These tools claim to be able to differentiate between human-written and AI-generated text with a very high accuracy, thereby ensuring the integrity of academic work. However, the use of these tools raised additional concerns, not only about their effectiveness (*Elkhatat, Elsaid & Almeer, 2023*; *Weber-Wulff et al., 2023*) but also their impact on equity (*Giray, 2024b*; *Liang et al., 2023*). AI text detection tools differ in their

design, precision, and underlying approaches. Similar to LLMs, these AI text detection tools operate as black box systems, lacking transparent explanations for their classifications of text as human-written or AI-generated, as they rely solely on pattern recognition. Moreover, even if these tools demonstrate high accuracy in detecting AI-generated text, none of them claim to be free of bias against certain groups of authors or writing styles.

Things become even more complicated since the involvement of LLMs in writing is not as straightforward as when the text is either purely human-written or entirely AI-generated. Increasingly, texts are hybrid or a mix of the two. For instance, humans may manually edit AI-generated text to varying degrees. Conversely, some researchers may write the first draft themselves and then run it through LLMs to improve readability. It is also possible for texts to undergo multiple iterations of AI and human editing, regardless of who authored the original draft. In these more nuanced cases, the final results can vary significantly, depending on several factors, including the quality of the original draft, the LLMs used, the prompt given to improve the text, and the extent of manual editing involved. This kind of hybrid text, also referred to as AI-assisted text, may cause detection tools to become less accurate, more biased, or both. This makes it more difficult to see the practical application of these tools in real-world scenarios without significant compromise.

To position this work within existing literature, previous studies have examined the performance of AI text detection tools primarily by comparing human-written and AI-generated texts, though most have not specifically addressed fairness or potential biases related to author background, such as native *vs.* non-native English speakers (*Perkins et al., 2024*; *Weber-Wulff et al., 2023*). Among the studies that do consider fairness, some are largely theoretical (*Giray, 2024a*, *2024b*), while another provides empirical evidence but is limited only to the comparison between human and AI-generated texts (*Liang et al., 2023*). Notably, none of them have yet empirically investigated fairness in detecting AI-assisted texts, a nuanced category situated between purely human-written and purely AI-generated content. This study addresses these gaps by first analyzing human *vs.* AI-generated text detection before extending the evaluation to include AI-assisted texts, with explicit consideration of potential biases against non-native authors as well as differences across academic disciplines. Additionally, this study employs the latest and most advanced large language models at the time (*i.e.*, ChatGPT o1 and Gemini 2.0 Pro Experimental) to ensure the findings reflect state-of-the-art capabilities and contemporary relevance.

Given the gaps identified above, this study aims to answer the following research questions:

RQ1. How accurate are AI text detection tools in identifying human-written, AI-generated, and AI-assisted texts in scholarly articles?

RQ2. Is there any accuracy-bias trade-off in AI text detection tools?

RQ3. Do certain groups of researchers face disadvantages when their work is evaluated by AI text detection tools?

By providing empirical evidence to answer these three questions, this study sheds light on the use and limitations of AI text detection tools, as well as their impact on fairness in

scholarly communication. In doing so, it contributes to a more nuanced understanding of both the opportunities and challenges presented by LLM involvement in scholarly publication.

The remainder of this article is organized as follows: the Materials & Methods section describes the experimental design, dataset selection, and methodologies used to evaluate AI text detection tools. The Results section presents findings from the two experimental scenarios. The Discussion section explores the accuracy-bias trade-offs, implications for fairness, practical recommendations, as well as study limitations and directions for future research. Finally, the Conclusion section summarizes the key insights from this study.

## MATERIALS AND METHODS

### Research design

This study is designed to evaluate the performance of AI text detection tools when applied to texts from scholarly journal articles. To provide a comprehensive analysis and to ensure a logical progression of findings, they will be examined for accuracy and fairness under two different experimental scenarios:

(1) Differentiating between original human-written texts and AI-generated texts, and
(2) Evaluating AI-assisted text, where the original human-written texts were run through LLMs to improve readability.

Considering factors such as length, accessibility, and content standardization, only the abstract from each scholarly article was used for evaluation.

### Dataset selection

The dataset for this research is a compilation of abstracts from peer-reviewed journal articles published by 2021, which is at least one year before the release of ChatGPT and other publicly available LLMs and their widespread use by researchers. To ensure balance in representation, the dataset was stratified based on disciplines (*i.e.*, technology & engineering, social sciences, and interdisciplinary) and authorship (*i.e.*, native and non-native English). Each discipline is represented by three journals: ACM Computing Surveys, IEEE Access, and PeerJ Computer Science for technology & engineering; Sociology, International Sociology, and SAGE Open for social sciences; and British Journal of Educational Technology, Computers & Education, and Education and Information Technologies for interdisciplinary.

Authors affiliated with institutions in the five Anglosphere countries (*i.e.*, Australia, Canada, New Zealand, United Kingdom, and United States) were chosen to represent the native English speakers. Meanwhile, to represent the non-native English speakers, only authors affiliated with institutions from countries where English is neither an official language nor spoken by more than half of the population were included. Furthermore, to avoid confounding variables, any articles with multiple authors from both native and non-native categories were excluded. The final dataset, as summarized in Table 1, comprises 72 articles with an equal distribution across disciplines and author categories.

**Table 1  Summary of dataset by discipline, journal, and country of affiliation.**

| Discipline | Journal | Native English | Non-Native English | Total articles (*n*) |
|---|---|---|---|---|
| Technology & Engineering | ACM Computing Surveys | Australia (*Mahmud, Ramamohanarao & Buyya, 2020*), | Chile (*Navarro & Rojas-Ledesma, 2020*), | 8 |
| | | UK (*Welsh & Benkhelifa, 2020*), | Jordan (*Khader & Al-Naymat, 2020*), | |
| | | US (*Wood, Najarian & Kahrobaei, 2020*; *Zave & Rexford, 2020*) | Mexico (*Falcón-Cardona & Coello, 2020*), | |
| | | | Peru (*Cornejo-Lupa et al., 2020*) | |
| | IEEE Access | Australia (*Christoe et al., 2021*), | Egypt (*Alshaer, Moawad & Ismail, 2021*), | 8 |
| | | New Zealand (*Kwon et al., 2021*), | Japan (*Hayashi, Shibanoki & Tsuji, 2021*), | |
| | | US (*Marino et al., 2021*; *Jacobs et al., 2021*) | South Korea (*Park et al., 2021*), | |
| | | | Taiwan (*Chen, 2021*) | |
| | PeerJ Computer Science | Canada (*Roussel, Achim & Auty, 2021*; *Bhat et al., 2021*), | Japan (*Fujita, 2021*), | 8 |
| | | UK (*Hudson & Moubayed, 2021*), | Russia (*Makarov et al., 2021*), | |
| | | US (*Bae, 2021*) | Sudan (*Elshoush, Al-Tayeb & Obeid, 2021*), | |
| | | | Vietnam (*Bui et al., 2021*) | |
| Social Sciences | International Sociology | Canada (*Coburn, 2021*), | Argentina (*Scribano, 2021*), | 8 |
| | | UK (*Holmes, 2021*), | China (*Jingting & Chao, 2021*), | |
| | | US (*Gallo-Cruz, 2021*; *Velitchkova, 2021*) | Guatemala (*Herrera & Rivera, 2021*), | |
| | | | South Korea (*Jung, 2021*) | |
| | SAGE Open | Australia (*James, Delfabbro & King, 2021*), | China (*Yen, Chen & Ho, 2021*), | 8 |
| | | Canada (*Wicklum et al., 2021*), | Japan (*Matsuo, 2021*), | |
| | | New Zealand (*Gibson et al., 2021*), | Oman (*Alhassan, 2021*), | |
| | | US (*Thomas & Cassady, 2021*) | Thailand (*Tharavanij, 2021*) | |
| | Sociology | Australia (*Neves & Mead, 2021*), | China (*Hu & Yin, 2021*), | 8 |
| | | UK (*Magrath, 2021*; *Brablec, 2021*), | Hungary (*Zakariás & Feischmidt, 2021*), | |
| | | US (*Murthy et al., 2021*) | Turkey (*Erkmen, 2021*; *Çelik, 2021*) | |
| Interdisciplinary | British Journal of Educational Technology | Australia (*Moro et al., 2021*), | China (*Li et al., 2021a*), | 8 |
| | | Canada (*Huang et al., 2021*), | Iran (*Latifi, Noroozi & Talaee, 2021*), | |
| | | UK (*Smith, 2021*), | Taiwan (*Hwang, Chien & Li, 2021*), | |
| | | US (*Staudt Willet & Carpenter, 2021*) | Turkey (*Unal & Uzun, 2021*) | |

| Discipline | Journal | Native English | Non-Native English | Total articles (*n*) |
|---|---|---|---|---|
| | Computers & Education | UK (*Herodotou et al., 2021*), | Brazil (*de Brito Lima, Lautert & Gomes, 2021*), | 8 |
| | | US (*Janakiraman et al., 2021*; *Wilson et al., 2021*; *Fletcher & Stanzione, 2021*) | China (*Li et al., 2021b*; *Le et al., 2021*), | |
| | | | South Korea (*Yang et al., 2021*) | |
| | Education and Information Technologies | Australia (*Turnbull, Chugh & Luck, 2021*), UK (*Hehir et al., 2021*), US (*Williams & Corwith, 2021*; *Kingsbury, 2021*)S | Indonesia (*Pratama, 2021*), Iraq (*Challob, 2021*), Saudi Arabia (*Altalhi, 2021*), Tunisia (*Pileh Roud & Hidri, 2021*) | 8 |
| Total | | 36 | 36 | 72 |

**Table 2 Standardized prompts used to produce AI-generated and AI-assisted abstracts.**

| Type | Prompt used |
|---|---|
| AI-Generated | Generate an abstract for each of these journal articles prepared for submission to the scientific journal "journal name": |
| | (1) Article 1 Title |
| | (2) Article 2 Title |
| | (3) … |
| | (4) Article 8 Title |
| AI-Assisted | Below are several research articles in preparation for submission to the "journal name" journal. Enhance the clarity and readability of the abstract but keep the title unchanged: |
| | (1) Title and Original Abstract of Article 1 |
| | (2) Title and Original Abstract of Article 2 |
| | (3) … |
| | (4) Title and Original Abstract of Article 8 |

## AI-generated and AI-assisted texts

For each original abstract in the dataset, its AI-generated version was produced using two of the most widely used LLMs available to the general public: OpenAI's ChatGPT and Google's Gemini. More specifically, only the most advanced versions available at the time of the experiment in late December 2024 were used: ChatGPT o1 and Gemini 2.0 Pro Experimental (Preview gemini-exp-1206). Note that for these AI-generated abstracts, only the article title and the journal name were provided to the LLMs as input; nothing from the original abstract was included. Subsequently, in a separate chat session, an AI-assisted abstract for each article was produced. To ensure consistency, a standardized prompt, as summarized in Table 2, was used to generate both AI-generated and AI-assisted abstracts, once for each journal.

This approach simulates a realistic and legitimate scenario where researchers use LLMs to refine their original writing, thus preserving the substance of the content while

**Table 3 Examples of the original, AI-generated, and AI-assisted abstracts from the same article.**

| Type | Text |
| --- | --- |
| Original Abstract (*Hwang, Chien & Li, 2021*) | Digital game-based learning (DGBL) is a frequently adopted mode for facilitating teaching and learning, where learners are immersed in game contexts to engage in gameplay and to construct meaningful knowledge. However, without guidance, students might have difficulties organizing what they have experienced in gaming contexts. Thus, in this research, a multidimensional repertory grid (MDRG) approach was proposed, and a digital game was implemented accordingly. Moreover, an experiment was executed in a geography course to examine the effectiveness of the strategy. A total of 83 high school students (15.5 years old on average) participated in the experiment. The experimental group learned with the multidimensional repertory grid-based educational game, while the control group learned with the conventional gaming mode. The analysis results revealed that the experimental group had better learning achievement as well as higher learning motivation, self-efficacy and metacognition awareness. In addition, the behavioral analysis and interview results revealed that those learning with the proposed strategy had a greater tendency to promote higher order thinking. Accordingly, the limitations and suggestions for future research are provided. |
| AI-Generated Chat GPT o1 | This study explores the use of a multidimensional repertory grid as a graphic organizer to systematically integrate digital games into classroom instruction, with the aim of enhancing students' learning performances and behaviors. The repertory grid framework aligns game elements, pedagogical strategies, and learning objectives, enabling teachers to develop coherent, learner-centered game-based activities. Employing a mixed-methods approach that includes classroom observations, student feedback, and performance assessments, the research examines the extent to which this visual tool supports effective instructional design. Findings show that the repertory grid can improve knowledge acquisition, foster positive learning attitudes, and sustain student motivation. Students engaged more deeply with content, demonstrated improved problem-solving skills, and exhibited greater collaboration. The organizer also guided teachers in selecting suitable games, scaffolding learning tasks, and making informed adjustments to instruction over time. These results highlight the potential of a multidimensional repertory grid to serve as a valuable resource for educators seeking to optimize the educational benefits of digital games, ultimately shaping more dynamic, impactful, and student-centered learning experiences. |
| AI-Generated Gemini 2.0 Pro Experimental | This study explores the efficacy of a multidimensional repertory grid (MDRG) as a graphic organizer in facilitating the implementation of digital games for enhanced student learning. MDRGs, based on Personal Construct Theory, enable the elicitation and representation of individuals' perceptions and understandings. We argue that MDRGs can help educators navigate the complexities of game-based learning by identifying key constructs related to learning outcomes and student behaviors. This research employs a mixed-methods approach, combining quantitative analysis of student performance data with qualitative analysis of MDRG constructs generated by both teachers and students. Findings indicate that using MDRGs as a planning and reflection tool promotes more intentional and targeted game integration. Furthermore, the process facilitated a shared understanding between teachers and students about the learning objectives and expected behaviors within the game-based environment. |
| AI-Assisted Chat GPT o1 | Digital game-based learning (DGBL) is widely adopted to facilitate teaching and learning by immersing students in game environments, allowing them to engage in gameplay and construct meaningful knowledge. However, without proper guidance, students may find it challenging to organize what they have learned in such contexts. To address this issue, this study proposes a multidimensional repertory grid (MDRG) approach and implements it within a digital game. An experiment was conducted in a geography course with 83 high school students (mean age = 15.5 years). The experimental group learned using the MDRG-based educational game, while the control group followed a conventional game-based approach. The results indicate that students in the experimental group outperformed their counterparts in learning achievement, motivation, self-efficacy and metacognitive awareness. Behavioral observations and interview data further reveal that students using the MDRG approach engaged in higher order thinking to a greater extent. Limitations and suggestions for future studies are also discussed. |
| AI-Assisted Gemini 2.0 Pro Experimental | Digital game-based learning (DGBL) offers an engaging approach to education, immersing learners in game contexts to foster meaningful knowledge construction. However, students may struggle to organize their experiences without adequate guidance. This research proposes a multidimensional repertory grid (MDRG) approach, integrated into a digital game, to address this challenge. An experiment was conducted with 83 high school students (average age 15.5 years) in a geography course to evaluate the effectiveness of this strategy. The experimental group used the MDRG-based educational game, while the control group engaged with a conventional gaming mode. Results showed that the experimental group achieved significantly better learning outcomes, alongside higher levels of learning motivation, self-efficacy, and metacognitive awareness. Furthermore, behavioral analysis and interviews revealed that students using the MDRG strategy exhibited a greater tendency towards higher-order thinking. The article concludes with limitations and suggestions for future research in this area. |

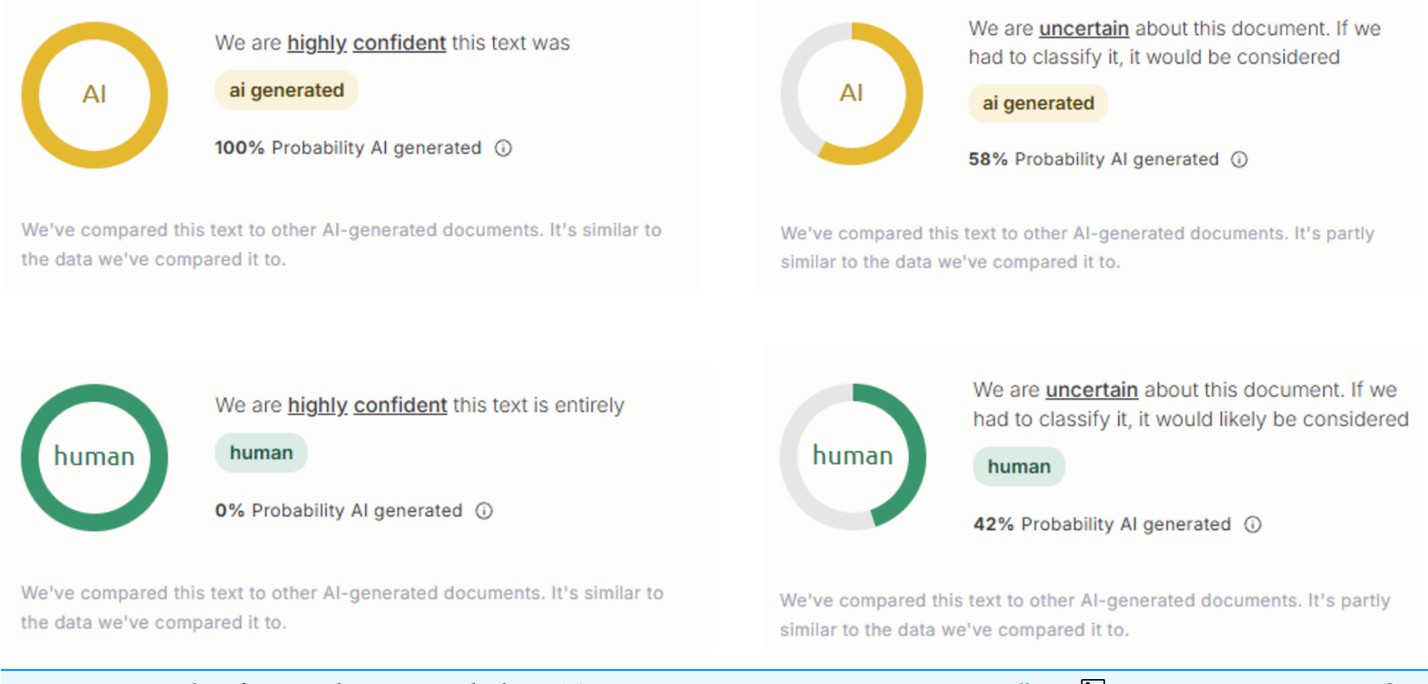

**Figure 1** Examples of AI text detection results by GPTZero.

improving its readability, rather than generating content from scratch. Table 3 provides an example of these abstracts in their original, AI-generated, and AI-assisted formats. These variations of abstracts represent the different types of text to be evaluated by AI text detection tools in the next step of the experiment. The complete dataset, along with the experimental results and Python code used to analyze and visualize the findings, is publicly available for download at the author's GitHub repository (https://github.com/ahmadrpratama/ai-text-detection-bias).

## AI text detection tools

In this study, three AI text detection tools that rank among the top results in Google search queries for "AI text detection tool" were included in the experiment: GPTZero, ZeroGPT, and DetectGPT. All three tools are freely available for public use, with optional premium subscriptions that unlock additional features such as longer word detection limits and detailed report generation. Each tool provides user with both the qualitative results (*i.e.*, categorical labels, such as "human," "AI," "mixed," or "uncertain") and quantitative results (*i.e.*, percentage scores representing the likelihood that a text is AI-generated), as compiled in Figs. 1–3.

## Evaluation metrics

In the first scenario, the task was to evaluate the tools' ability to distinguish between original human-written and AI-generated abstracts. Ground truth is clearly established in this scenario, as the origin of each text (human or AI-generated) is

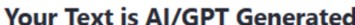

**Your Text is AI/GPT Generated**

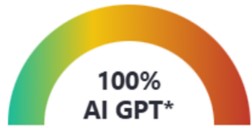

100%
AI GPT*

**Most of Your Text is AI/GPT Generated**

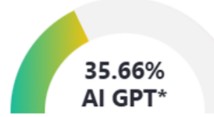

35.66%
AI GPT*

**Your Text is Human written**

0%
AI GPT*

**Your Text is Most Likely Human written, may include parts generated by AI/GPT**

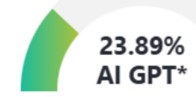

23.89%
AI GPT*

**Figure 2 Examples of AI text detection results by ZeroGPT.**

**Classification**
We are highly confident this text has been entirely written by AI

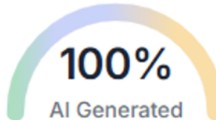

100%
AI Generated

**Classification**
Most of your text has been written by AI

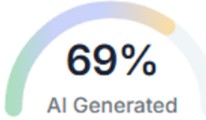

69%
AI Generated

**Classification**
We are highly confident this text has been entirely written by human

0%
AI Generated

**Classification**
We are confident this text has been written by human

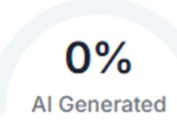

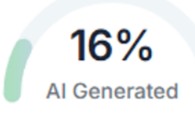

16%
AI Generated

**Figure 3 Examples of AI text detection results by DetectGPT.**

explicitly known. This allows for precise evaluation of the tools' performance using the following metrics:

1. Accuracy: The percentage of correctly classified abstracts.
2. False positive rate (FPR): The percentage of original abstracts where the tool failed to classify as human written.

3. False negative rate (FNR): The percentage of AI-generated abstracts that the tool failed to classify as AI-generated.

In addition, two complementary metrics were used to capture broader patterns of false positives across the dataset, in which original abstracts were misclassified as AI-generated. These two metrics offer a better picture of how such tools could potentially harm authors:

4. False accusation rate (FAR): The percentage of original abstracts with at least one false positive result.
5. Majority false accusation rate (MFAR): The percentage of original abstracts with more false positive results than true negative results.

The second scenario is more nuanced, as the task involved evaluating AI-assisted abstracts, which are human-authored texts enhanced by LLMs. Unlike the first scenario, determining ground truth for AI-assisted texts is much more complex because these texts blend both human and AI contributions. In this case, the focus shifted to quantitative analysis, evaluating the percentage scores provided by the tools. The analysis involved:

6. Summary statistics of scores, which include min, max, quartiles, median, mean, and standard deviation.
7. Statistical tests to identify significant differences in detection scores across disciplines (*i.e.*, technology & engineering, social sciences, interdisciplinary), author categories (*i.e.*, native, non-native), and LLMs (*i.e.*, ChatGPT o1, Gemini 2.0 Pro Experimental).

Significant differences in scores would indicate potential biases against certain groups of authors, such as non-native speakers or researchers in specific disciplines. The purpose of this scenario was to highlight the challenge of accurately evaluating partially AI-generated content, where no clear-cut ground truth exists.

Nevertheless, considering that in many cases, these AI text detection tools do make extreme classifications (*i.e.*, 0% or 100%), even for AI-assisted texts, their performance was further evaluated with two additional metrics:

8. Under-Detection Rate (UDR): The percentage of AI-assisted abstracts labeled as 0% AI by the tool, indicating a failure to detect any AI contribution.
9. Over-Detection Rate (ODR): The percentage of AI-assisted abstracts labeled as 100% AI by the tool, incorrectly attributing the entire text to AI and disregarding human contribution.

## RESULTS

### Scenario 1: Original *vs.* AI-generated abstracts

The performance of all three AI text detection tools in classifying abstracts, where the ground truth is clearly established as either human or AI-generated, is summarized in Table 4. The five metrics used in this scenario were analyzed across author categories and disciplines to identify patterns of reliability and potential bias.

As shown in Table 4, out of the three AI text detection tools in this study, GPTZero achieved the highest accuracy at 97.22% with a 0% FPR and a very low FNR at 2.78%.
**Table 4 Overall performance metrics from Scenario 1: Original *vs.* AI-generated abstracts.**

| Metric | GPTZero (%) | ZeroGPT (%) | DetectGPT (%) |
|---|---|---|---|
| Accuracy | 97.22 | 64.35 | 54.63 |
|   - By author categories | | | |
|     - Native | 99.07 | 64.81 | 58.33 |
|     - Non-Native | 97.22 | 63.89 | 50.93 |
|   - By disciplines | | | |
|     - Technology & Engineering | 100.00 | 61.11 | 56.94 |
|     - Social Sciences | 98.61 | 66.67 | 63.89 |
|     - Interdisciplinary | 95.83 | 65.28 | 43.06 |
| | | | |
| False Positive Rate (FPR) | 0.00 | 16.67 | 31.94 |
|   - By author categories | | | |
|     - Native | 0.00 | 19.44 | 27.78 |
|     - Non-Native | 0.00 | 13.89 | 36.11 |
|   - By disciplines | | | |
|     - Technology & Engineering | 0.00 | 12.50 | 41.67 |
|     - Social Sciences | 0.00 | 12.50 | 12.50 |
|     - Interdisciplinary | 0.00 | 25.00 | 41.67 |
| | | | |
| False Negative Rate (FNR) | 2.78 | 45.15 | 52.08 |
|   - By author categories | | | |
|     - Native | 2.78 | 43.06 | 48.61 |
|     - Non-Native | 8.33 | 47.22 | 55.56 |
|   - By disciplines | | | |
|     - Technology & Engineering | 2.08 | 52.08 | 43.75 |
|     - Social Sciences | 4.17 | 43.75 | 47.92 |
|     - Interdisciplinary | 2.08 | 39.58 | 64.58 |
| | | | |
| False Accusation Rate (FAR) | | | 44.44 |
|   - By author categories | | | |
|     - Native | | | 44.44 |
|     - Non-Native | | | 44.44 |
|   - By disciplines | | | |
|     - Technology & Engineering | | | 45.83 |
|     - Social Sciences | | | 25.00 |
|     - Interdisciplinary | | | 62.50 |
| | | | |
| Majority False Accusation Rate (MFAR) | | | 4.17 |
|   - By author categories | | | |
|     - Native | | | 2.78 |
|     - Non-Native | | | 5.56 |

*(Continued)*

| Metric | GPTZero (%) | ZeroGPT (%) | DetectGPT (%) |
|---|---|---|---|
| - By disciplines | | | |
|     - Technology & Engineering | | | 8.33 |
|     - Social Sciences | | | 0.00 |
|     - Interdisciplinary | | | 4.17 |

These numbers suggest strong reliability of GPTZero in differentiating between human-written and AI-generated texts in this first scenario, where the ground truth is clearly established. ZeroGPT performed much worse, with an overall accuracy of just 64.35%, a relatively high FPR at 16.67%, and even higher FNR at 45.14%. DetectGPT, on the other hand, despite claiming a 99% accuracy rate, performed the worst in practice, achieving merely 54.63% accuracy. This makes it virtually no better than random guessing. It also exhibited much higher FPR and FNR values at 31.94% and 52.08%, respectively.

In addition to the three tool-specific metrics above, the other two dataset-wide metrics (*i.e.*, FAR and MFAR) further highlight the broader risks of misclassification that could potentially harm researchers by falsely accusing them of using AI for their original writings. The FAR of 44.44% means nearly half of the original abstracts in the dataset were misclassified as AI-generated by at least one AI text detection tool, which is alarming. The MFAR, while notably much lower at 4.17%, still indicates the potential risk of consensus false accusations by multiple AI text detection tools, albeit less frequently.

When examining variability across author categories and disciplines, slight differences in accuracy were observed. All three tools performed slightly more accurately for native authors compared to non-native authors. However, performance consistency for FPR and FNR across tools and categories was lacking. For instance, ZeroGPT exhibited a higher FPR for original abstracts written by native authors compared to those written by non-native authors, whereas DetectGPT demonstrated the opposite trend. Regarding discipline-based variability, FNR values for ZeroGPT and DetectGPT were lowest for social sciences abstracts, but their highest FNRs occurred in different categories: ZeroGPT struggled the most with AI-generated abstracts in technology & engineering disciplines, while DetectGPT performed worst with AI-generated abstracts in interdisciplinary disciplines.

For the dataset-wide metrics, FAR showed no difference between native and non-native authors, remaining constant at 44.44%. However, MFAR was slightly higher for non-native authors (5.56%) compared to native authors (2.78%). Across disciplines, social sciences abstracts consistently exhibited the lowest FAR (25%) and MFAR (0%), whereas interdisciplinary abstracts had the highest FAR (62.50%), and technology & engineering abstracts showed the highest MFAR (8.33%). These findings underscore the tools' struggles with texts in interdisciplinary and technology & engineering disciplines, particularly those authored by non-native English speakers.

While the results from this first scenario offer some insight into how these tools perform, they only address cases where the ground truth is clear. In real-world contexts these days, texts often blend human-written texts with AI-generated texts. The second

**Table 5 Summary statistics of detection scores from Scenario 2: AI-assisted abstracts.**

| Statistic | GPTZero (%) | ZeroGPT (%) | DetectGPT (%) |
|---|---|---|---|
| Mean (SD) | 37.65 (39.99) | 20.92 (29.24) | 52.36 (47.40) |
| • By author categories | | | |
| • Native | 30.68 (35.29) | 21.91 (27.79) | 54.40 (46.79) |
| • Non-native | 44.61 (43.33) | 19.94 (30.79) | 50.32 (48.24) |
| • By disciplines | | | |
| • Tech & Engineering | 41.04 (40.21) | 10.99 (21.30) | 53.90 (48.41) |
| • Social Sciences | 35.21 (38.12) | 20.06 (33.15) | 52.10 (46.92) |
| • Interdisciplinary | 36.69 (42.16) | 31.73 (28.77) | 51.08 (47.81) |
| • By LLMs | | | |
| • ChatGPT o1 | 19.79 (30.51) | 10.04 (21.50) | 29.47 (21.50) |
| • Gemini 2.0 Pro | 55.50 (40.55) | 31.80 (31.93) | 75.25 (31.93) |
| | | | |
| Q1, Median, Q3 | 2.00, 12.50, 80.25 | 0.00, 0.00, 40.51 | 0.00, 80.50, 100.00 |
| • By author categories | | | |
| • Native | 3.00, 9.50, 58.00 | 0.00, 0.00, 44.84 | 0.00, 81.50, 100.00 |
| • Non-native | 2.00, 22.50, 99.25 | 0.00, 0.00, 36.38 | 0.00, 80.50, 100.00 |
| • By disciplines | | | |
| • Tech & Engineering | 2.00, 26.50, 80.25 | 0.00, 0.00, 5.85 | 0.00, 80.50, 100.00 |
| • Social Sciences | 2.00, 14.00, 74.25 | 0.00, 0.00, 34.72 | 0.00, 79.50, 100.00 |
| • Interdisciplinary | 3.00, 9.00, 90.25 | 0.00, 30.48, 52.96 | 0.00, 83.50, 100.00 |
| • By LLMs | | | |
| • ChatGPT o1 | 1.00, 5.00, 25.75 | 0.00, 0.00, 0.00 | 0.00, 0.00, 0.00 |
| • Gemini 2.0 Pro | 9.00, 58.00, 100.00 | 0.00, 29.34, 53.06 | 0.00, 78.50, 91.00 |
| | | | |
| Min, Max | 0.00, 100.00 | 0.00, 100.00 | 0.00, 100.00 |
| • By author categories | | | |
| • Native | 0.00, 100.00 | 0.00, 100.00 | 0.00, 100.00 |
| • Non-native | 0.00, 100.00 | 0.00, 100.00 | 0.00, 100.00 |
| • By disciplines | | | |
| • Tech & Engineering | 0.00, 100.00 | 0.00, 79.43 | 0.00, 100.00 |
| • Social Sciences | 0.00, 100.00 | 0.00, 100.00 | 0.00, 100.00 |
| • Interdisciplinary | 0.00, 100.00 | 0.00, 96.64 | 0.00, 100.00 |
| • By LLMs | | | |
| • ChatGPT o1 | 0.00, 100.00 | 0.00, 100.00 | 0.00, 100.00 |
| • Gemini 2.0 Pro | 0.00, 100.00 | 0.00, 100.00 | 0.00, 100.00 |

scenario fills this gap by examining how these tools handle AI-assisted abstracts and whether biases emerge across author statuses or disciplines.

## Scenario 2: AI-assisted abstracts

In this scenario, quantitative measures (*i.e.*, the probability scores of AI-generated text) are used to assess the performance of each AI text detection tool, and the summary statistics are

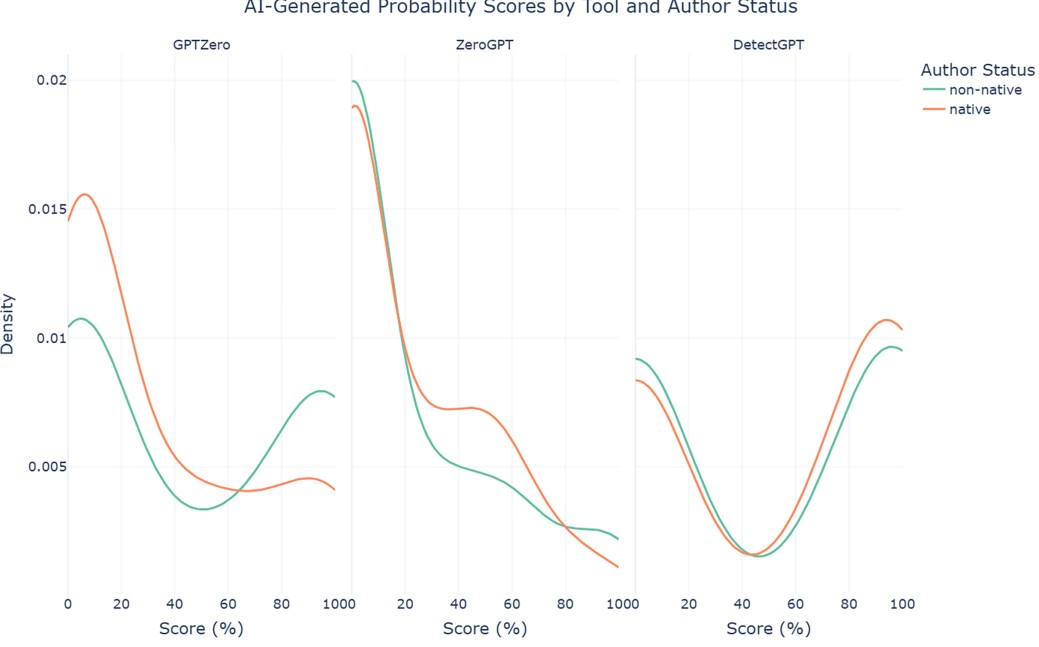

**Figure 4** Density plots of AI-generated probability scores for AI-assisted abstracts from each AI text detection tool by author status.

presented in Table 5. This approach is more suitable given that the evaluated texts (*i.e.*, AI-assisted abstracts) blend human-written content with AI-generated enhancements, creating a more nuanced category where clear-cut labels are no longer applicable.

The findings reveal a clear bias in GPTZero, which tends to assign lower AI-generated probabilities to AI-assisted abstracts written by native authors (median = 9.50%, mean = 30.68%, SD = 35.29%) and higher probabilities to those written by non-native authors (median = 22.50%, mean = 44.61%, SD = 43.33%). Welch's t-test confirms this disparity, showing a statistically significant difference (t = −2.115, $p$ = 0.036). This pattern, visualized in the density plot in Fig. 4, highlights a concerning imbalance that potentially places non-native authors at a serious disadvantage, amplifying existing inequities in scholarly publishing. In contrast, the other two tools, ZeroGPT and DetectGPT, exhibit more balanced performance, as evidenced by their Welch's t-test results, which show no statistically significant differences ($p$ = 0.687 and $p$ = 0.607, respectively). However, given GPTZero's high accuracy in the previous scenario, these findings raise significant concerns about the unintended consequences of its use.

The findings also reveal that ZeroGPT struggles significantly in detecting AI-assisted abstracts from technology & engineering disciplines (median = 0%, mean = 10.99%, SD = 21.30%), compared to social sciences (median = 0%, mean = 20.06%, SD = 33.15%) and even more so interdisciplinary abstracts (median = 30.48%, mean = 31.73%, SD = 28.77%). Welch's ANOVA confirms this disparity, with a statistically significant difference (F = 7.94, $p$ < 0.001). This pattern is clearly illustrated in the density plot in Fig. 5. GPTZero (F = 0.28, $p$ = 0.759) and DetectGPT (F = 0.04, $p$ = 0.960), on the other

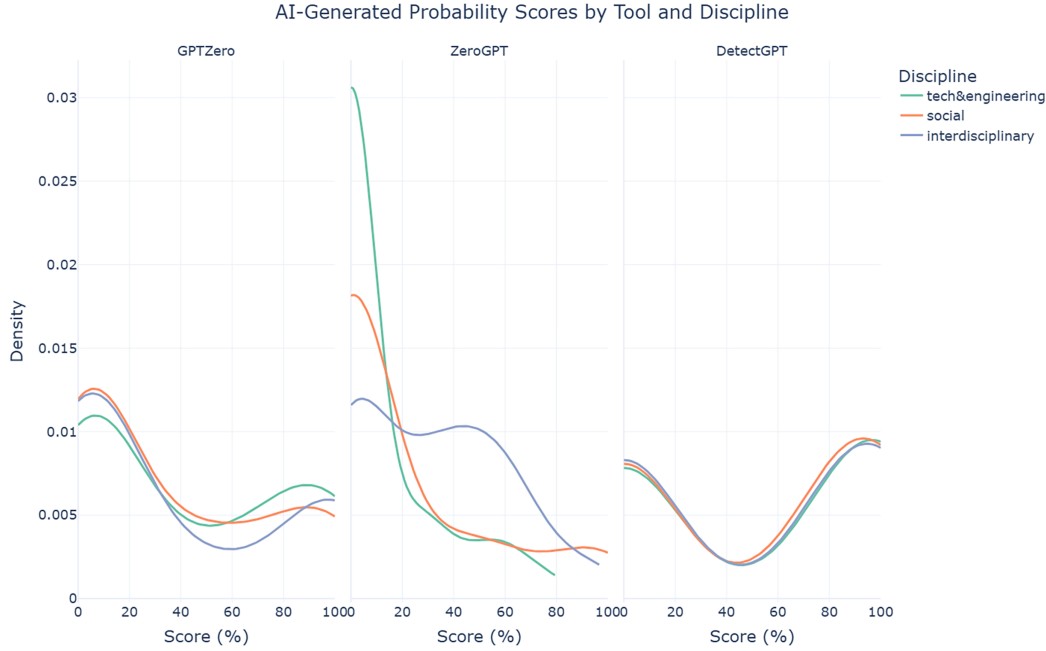

**Figure 5 Density plots of AI-generated probability scores for AI-assisted abstracts from each AI text detection tool by discipline.**

hand, exhibit relatively more consistent performance across disciplines, as reflected in their non-significant Welch's ANOVA results. These findings highlight the challenges AI text detection tools face when applied to different disciplines, with ZeroGPT showing particular weaknesses in more technical fields.

In terms of LLMs, the findings reveal that abstracts enhanced by Gemini 2.0 Pro Experimental are significantly more likely to be detected as AI-generated than those enhanced by ChatGPT o1, across all three detector tools. GPTZero assigns a notably higher mean probability score to Gemini-enhanced abstracts (mean = 55.50%, SD = 40.55%) compared to those enhanced by ChatGPT (mean = 19.79%, SD = 30.51%). Welch's t-test indicates this difference is statistically significant ($t = -5.97$, $p < 0.001$). Similarly, ZeroGPT assigns higher mean probability scores to Gemini-enhanced abstracts (mean = 31.80%, SD = 31.93%) compared to ChatGPT-enhanced abstracts (mean = 10.04%, SD = 21.50%), with Welch's t-test confirming significance ($t = -4.80$, $p < 0.001$). DetectGPT also shows the same pattern, assigning much higher mean probability scores to Gemini-enhanced abstracts (mean = 75.25%, SD = 31.93%) compared to ChatGPT-enhanced abstracts (mean = 29.47%, SD = 21.50%), a huge difference that once again confirmed to be statistically significant with Welch's t-test ($t = -6.60$, $p < 0.001$). These findings indicate that using either LLM to enhance text increases the AI-generated probability scores assigned by detection tools, even when the underlying content is human-authored. However, among the two models used to enhance the original text, the risk of elevated scores appears to be more pronounced with Gemini-enhanced text, as depicted in the density plot in Fig. 6. This may reflect distinct stylistic or linguistic patterns detectable by current tools.

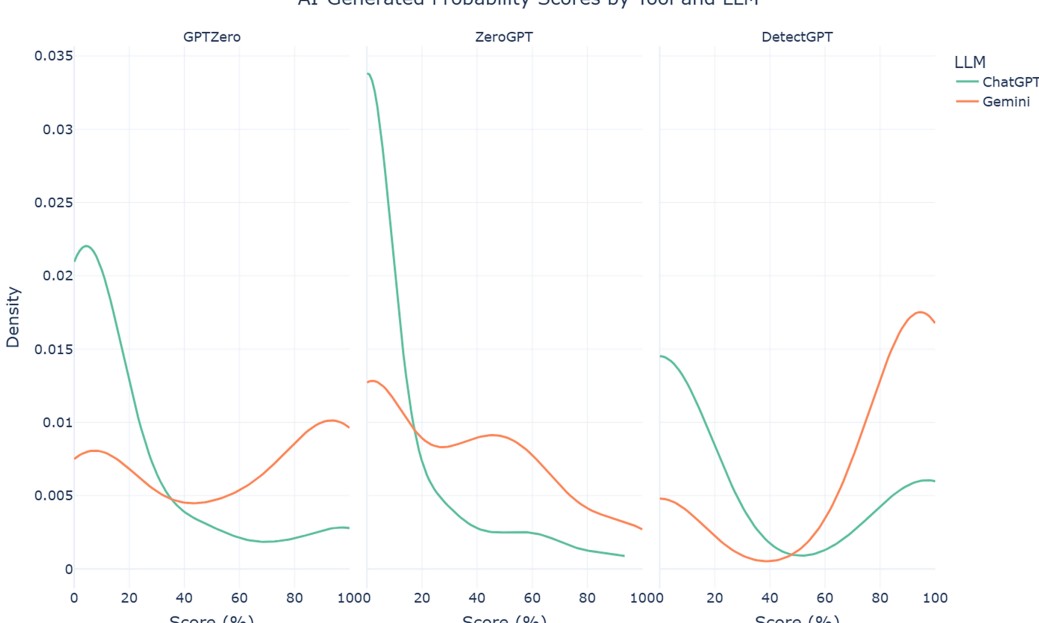

**Figure 6** Density plots of AI-generated probability scores for AI-assisted abstracts from each AI text detection tool by LLM.

**Table 6 Overall performance metrics from Scenario 2: AI-assisted abstracts.**

| Metric | GPTZero (%) | ZeroGPT (%) | DetectGPT (%) |
|---|---|---|---|
| Under-Detection Rate (UDR) | 3.47 | 58.33 | 44.44 |
| • By author categories | | | |
| • Native | 4.17 | 52.78 | 41.67 |
| • Non-Native | 2.78 | 63.89 | 47.22 |
| • By disciplines | | | |
| • Technology & Engineering | 4.17 | 75.00 | 43.75 |
| • Social Sciences | 4.17 | 66.67 | 43.75 |
| • Interdisciplinary | 2.08 | 33.33 | 45.83 |
| • By LLMs | | | |
| • ChatGPT o1 | 5.56 | 76.39 | 69.44 |
| • Gemini 2.0 Pro | 1.39 | 40.28 | 19.44 |
| | | | |
| Over-Detection Rate (ODR) | 18.06 | 2.08 | 34.03 |
| • By author categories | | | |
| • Native | 11.11 | 1.39 | 33.33 |
| • Non-Native | 25.00 | 2.78 | 34.72 |
| • By disciplines | | | |
| • Technology & Engineering | 16.67 | 0.00 | 41.67 |
| • Social Sciences | 12.50 | 6.25 | 29.17 |
| • Interdisciplinary | 25.00 | 0.00 | 31.25 |

| Table 6 (continued) | | | |
|---|---|---|---|
| Metric | GPTZero (%) | ZeroGPT (%) | DetectGPT (%) |
| • By LLMs | | | |
| • ChatGPT o1 | 8.33 | 0.00 | 22.22 |
| • Gemini 2.0 Pro | 27.78 | 4.17 | 45.83 |

Consequently, authors relying on Gemini Pro 2.0 Experimental are at greater risk of their work being unfairly flagged as AI-generated by detection tools than those relying on ChatGPT o1.

Finally, Table 6 presents UDR and ODR for AI-assisted abstracts, broken down by author status, discipline, and LLM. Across all tools, the ODR for non-native authors is consistently higher than for native authors, with GPTZero showing the most pronounced disparity: its ODR for non-native authors (25%) is more than double that of native authors (11%). In other words, one in four non-native authors who use AI to help refine their text is at risk of being accused of having submitted an entirely AI-generated text by GPTZero, while the risk for native authors is closer to one in ten. This pattern, once again, underscores the disproportionate risks faced by non-native authors, who ironically are the ones most likely to benefit from using LLMs to enhance their English writing, yet are also the most likely to have their contributions dismissed entirely by detection tools.

Meanwhile, although ZeroGPT seemed to be the most prudent, with the lowest ODR value of all three tools, it also exhibited the highest UDR, with values consistently exceeding 50% across all author statuses and disciplines except for interdisciplinary. Furthermore, while other tools typically use a 50% cutoff to classify text as AI-generated, ZeroGPT relies on a lower, less transparent threshold. For instance, it labeled some original abstracts in Scenario 1 as AI-generated even though their scores were below 30%. This combination of moderately high FPR in Scenario 1 and high UDR in Scenario 2 highlights the tool's difficulty in balancing false positives and false negatives across varying contexts.

On the other hand, DetectGPT showed moderately high values for both UDR and ODR, indicating a strong tendency to make bold yet incorrect classifications. Coupled with the lowest accuracy and the highest FPR and FNR observed in the previous scenario, these findings underscore DetectGPT's limitations. Ultimately, DetectGPT emerged as the worst-performing AI text detection tool in this study.

## DISCUSSION

The findings in this study highlight significant challenges in the use of AI text detection tools, driven in part by the rapidly evolving landscape of LLMs and generative AI. Updates to LLMs can quickly render existing detection tools ineffective, as demonstrated by the poor performance of ZeroGPT and DetectGPT in this study. These tools struggled to handle texts generated by ChatGPT o1 and Gemini 2.0 Pro Experimental, the two most advanced models available at the time of the study, revealing limitations in their adaptability. These findings are consistent with previous research evaluating AI text detection tools on earlier versions of these LLMs (*Perkins et al., 2024*;

*Elkhatat, Elsaid & Almeer, 2023*; *Weber-Wulff et al., 2023*; *Liang et al., 2023*), suggesting that the detection-based approach may be fundamentally flawed. This dynamic creates a perpetual cycle of advancement and adaptation, where detection tools must continuously try to catch up. However, this urgency often leads to trade-offs, such as introducing biases against specific groups of authors or disciplines, as seen with GPTZero, or increasing the risk of false accusations that disproportionately affect non-native authors.

These findings have significant implications for scholarly publication. Non-native authors, who already face systemic challenges, are disproportionately affected by the biases of these tools, particularly GPTZero. The risk of being falsely accused of submitting AI-generated content, even when AI was used solely to improve the readability of their own writing, as shown in Scenario 2, could further exacerbate existing inequities in access to publication opportunities. This is especially concerning given that non-native authors are far more likely than native authors to rely on AI tools to enhance the clarity and readability of their work (*Liao et al., 2024*). The ethical implications of using AI text detection tools go beyond technical performance. The lack of transparency in how these tools evaluate text, combined with their potential biases, poses serious risks to authors' reputations and careers. Institutions, journal editors, and publishers must take these risks into account when adopting such tools, ensuring they are used as supplementary aids rather than definitive decision-makers.

Given the widely acknowledged language barriers in scholarly publication, LLMs have the potential to level the playing field for non-native authors. Rather than stigmatizing their use, it may be more beneficial to embrace these tools as valuable aids for improving the clarity and accessibility of academic writing. Notably, some researchers, journal editors, and even publishers have voiced support for the responsible adoption of LLMs, recognizing their potential to enhance inclusivity and reduce linguistic barriers (*Seghier, 2023*; *Kaebnick et al., 2023*; *Koller et al., 2024*). The focus, therefore, should not be on penalizing authors who use LLMs, but on promoting ethical use, ensuring transparency, and holding authors accountable for the accuracy and integrity of their work.

At the end of the day, LLMs are merely tools, and like any tool, their misuse or overreliance without oversight can lead to significant issues. One of the most critical concerns in content generated by LLMs is AI hallucinations (*Salvagno, Taccone & Gerli, 2023*; *Athaluri et al., 2023*; *Hatem et al., 2023*), where LLMs generate false or misleading information, essentially fabrications and falsifications (*Emsley, 2023*). Another potential issue is plagiarism, where human authors use a simple prompt to have the LLM generate text from scratch and then present it as their own original work. These behaviors constitute scientific misconduct and are unacceptable. Authors who fail to verify and validate any AI involvement in the production of their content and manuscript preparation should be held accountable for the consequences. Ultimately, ethical use and transparency are essential for the responsible integration of LLMs into scholarly publication.

Furthermore, a shift in the default assumption may be necessary to achieve greater transparency. Currently, many journals and publishers require authors to disclose the use of LLMs for improving readability (*Cheng & Wu, 2024*; *Ganjavi et al., 2024*;

*Flanagin et al., 2023*). Given the widespread availability of LLMs, their increasing integration into writing workflows, and evidence that some authors choose not to disclose their AI use (*Pesante, Mauffrey & Parry, 2024*; *Chemaya & Martin, 2024*), it may be more practical to assume that every submitted manuscript has had some level of AI involvement for copyediting or improving readability, unless authors explicitly state otherwise. For instance, requiring a disclaimer such as "no AI involvement" could serve as a clear affirmation that the manuscript was entirely written and edited by humans without any AI support. That said, any other forms of LLM involvement, such as generating text, images, or other content from scratch using a prompt, should be fully disclosed. Ultimately, the emphasis should remain on the substance of the research and the scientific rigor it reflects, rather than the tools used in its production.

## Limitations and future work

Several limitations should be acknowledged to appropriately contextualize the results of this study. First, the selection of articles published up to 2021, while intentionally chosen to predate the public accessibility of generative AI tools like ChatGPT, still overlaps with earlier GPT-based technologies (*e.g.*, GPT-2 released in 2019 and GPT-3 released in 2020). Although fully generative tools capable of producing coherent, lengthy texts from scratch were not widely available to the general public until late 2022, earlier GPT-based tools (*e.g.*, Grammarly) were primarily used for language refinement rather than text generation. Therefore, it remains reasonable to assume that the original texts in journals published in 2021 were human-authored, though the possibility that GPT-based tools contributed to editing or polishing the writing cannot be entirely ruled out.

Second, the disciplinary scope of the selected journals was intentionally limited due to practical constraints. This study included journals from technology & engineering and social sciences to represent contrasting fields, as well as interdisciplinary journals positioned in between. However, this limited representation may not fully generalize to broader STEM and non-STEM categories. Future research should expand disciplinary coverage to include a broader spectrum of fields, such as natural sciences, humanities, health sciences, and professional disciplines like medicine, law, or economics to strengthen the generalizability of the results. Expanding the range of disciplines would not only improve representativeness but also support the development of a larger and more diverse dataset, enabling more robust and nuanced analyses across fields.

Finally, the analysis focused on a limited number of popular and publicly available AI detection tools. Considering the rapid evolution of AI detection technologies, future studies should incorporate additional detection tools, particularly those commonly used by higher education institutions (*e.g.*, Turnitin) to enhance the real-world applicability and relevance of the findings.

Addressing these limitations in future work will further enrich our understanding of AI-generated and AI-assisted writing, highlight the inherent flaws and biases of AI text detection tools, and reinforce the need for alternative approaches that prioritize transparency, fairness, and ethical use of AI in scholarly publication.

## CONCLUSION

This study evaluates the performance of popular AI text detection tools and highlights the challenges associated with their practical use in the scholarly publication realm. Despite claims of high accuracy, most tools in this study struggled with both accuracy and consistency when applied to texts involving the latest versions of LLMs. The study also reveals notable accuracy-bias trade-offs, where tools with higher overall accuracy (such as GPTZero) exhibit stronger biases against non-native authors and certain disciplines, in both original and AI-assisted human writings.

The findings underscore the need for promoting ethical and transparent use of LLMs over detection-focused approaches. When used responsibly, LLMs have the potential to help level the playing field for non-native authors. Clear guidelines that encourage disclosure of AI involvement while maintaining author accountability are essential to fostering trust in scholarly publishing. As LLMs continue to evolve, collaborative efforts are needed to ensure that scholarly communication remains equitable by keeping the focus on the quality and integrity of the research, rather than on the tools used in its production.

## ACKNOWLEDGEMENTS

In addition to using ChatGPT and Gemini to generate part of the dataset in this study, the author also used ChatGPT to improve the readability of this manuscript and Gemini in Google Colab to troubleshoot and fix errors in the Python code used for data analysis and visualization.

### Funding
The author received no funding for this work.

### Competing Interests
The authors declare that they have no competing interests.

### Author Contributions
- Ahmad R. Pratama conceived and designed the experiments, performed the experiments, analyzed the data, performed the computation work, prepared figures and/or tables, authored or reviewed drafts of the article, and approved the final draft.

### Data Availability
The complete dataset, along with the experimental results and Python code used to analyze and visualize the findings, is available at GitHub and Zenodo:

https://github.com/ahmadrpratama/ai-text-detection-bias

ahmadrpratama. (2025). ahmadrpratama/ai-text-detection-bias: v1.0.0: Data and code for "The Accuracy-Bias Trade-Offs in AI Text Detection Tools and Their Impact on Fairness in Scholarly Publication" (PeerJ Computer Science) (v1.0.0). Zenodo. https://doi.org/10.5281/zenodo.15490003.
## Supplemental Information

Supplemental information for this article can be found online at http://dx.doi.org/10.7717/peerj-cs.2953#supplemental-information.

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
