# Peer review of "The accuracy-bias trade-offs in AI text detection tools and their impact on fairness in scholarly publication"

_PeerJ Computer Science, doi:10.7717/peerj-cs.2953_

## Round 0.1 · original submission · Major Revisions

Reviewers have commented about your manuscript and you would notice that they are advising for a revision. Please note that these revisions are towards dataset chosen, presentation and structure of the content, etc. Hence while revising the manuscript please make a list of corrections against each of the comments.

**Language Note:** The review process has identified that the English language must be improved. PeerJ can provide language editing services - please contact us at [email protected] for pricing (be sure to provide your manuscript number and title). Alternatively, you should make your own arrangements to improve the language quality and provide details in your response letter. – PeerJ Staff

Reviewer 1 ·

Basic reporting

The study contributes to the discussion on the suitability of AI-generated text detectors in scholarly publishing. Overall, the experiment is well-designed. The results are in line with previous findings yet bring new information, especially when it comes to potential disadvantages to non-native English-speaking authors. I enjoyed reading the paper and certainly recommend it for publication.

There are several issues that -- in my view -- influence the interpretability and generalizability of the results, and should be addressed. See my further comments.

Experimental design

You selected the studies published before 2021 as certainly human-written. Even though ChatGPT did not exist at that time, GPT-based tools were already available (GPT-2 was released in 2019). Moreover, tools like Grammarly existed at that time. To ensure that no LLMs based on transformer technology influenced the content in any way, you should select the papers published before 2017 (publication of the paper Attention is all you need). I understand you can't change the 2021 cutoff, and I don't think it influenced the results. Nonetheless, I believe it is worth mentioning as a study limitation.

The choice of the fields of study is very limited. For STEM, you consider only computer science journals and IEEE Access. Natural sciences and math are not represented at all. I suggest you don't talk about STEM but about "technology" or "technology and engineering". The same issue applies to non-STEM, which is basically limited to sociology only. Again, the terminology in the paper should be changed accordingly.

Validity of the findings

Were there any differences between the generators used? You present the results for STEM/non-STEM and native/non-native. It would be great to see if the generator's choice influences the results. If there are statistically significant differences, please present them in the tables. If the results are not significantly different, just comment on this fact in the text.

You claim that "The complete dataset, along with the experimental results and Python code used to analyze and visualize the findings, is publicly available for download at the author's GitHub page."
Unfortunately, there is no direct link, and I could not find such a repository in GitHub. Therefore, I was unable to check the raw data. This is the reason why I am suggesting "MAJOR REVISION"; I am convinced that one more round of review is necessary.

Additional comments

Thank you for the opportunity to read the paper. After addressing the issues mentioned above, it will be a great paper worth reading.

·

Basic reporting

The paper is well-written, clear, and easy to read. The language is professional, and the terminology is appropriately used. The authors cite a wide range of sources, effectively providing the necessary background for their research.

The paper is well-structured, with figures and tables that enhance the reader’s understanding of the findings and research methodology. The visuals are detailed and contribute meaningfully to the presentation of the results. Additionally, the authors provide sufficient detail on the metrics calculated and their interpretation.

Experimental design

The research aligns well with the aims and scope of the journal. The research questions are clearly formulated in the Introduction and are systematically addressed through experiments with AI text detectors. The methodology is well-documented, with a clear explanation of the experimental design and the metrics used.

Validity of the findings

The study demonstrates strong research validity.

Additional comments

- The abbreviation AI appears in the abstract without first introducing the full term. It is recommended to use the full term (artificial intelligence) in the abstract and introduce the abbreviation in the main text.
- The capitalization of "artificial intelligence" and "large language models" in the introduction is unnecessary. Common nouns and descriptive terms should be in lowercase, while proper nouns and specific model names (e.g., GPT-4, BERT, T5) should be capitalized.
- Abbreviations should be used consistently throughout the paper. Standard practice is to provide the full term at first mention, followed by the abbreviation in parentheses (e.g., "artificial intelligence (AI)"), after which only the abbreviation should be used. The full terms for UDR and ODR are used again after their abbreviations were introduced—this should be corrected.
- The Introduction should conclude with a brief description of the paper’s structure, summarizing its sections.
- The paper lacks a Related Work section. To the best of the reviewer’s knowledge, prior research has already explored the distinction between AI-generated, AI-assisted, and human-written content for both native and non-native speakers. The authors should discuss these studies and position their work within the existing body of knowledge.
- A discussion of the study's limitations should be included. This will improve transparency and help contextualize the findings.
- The first entry in the References section is incomplete and lacks bibliographical details. This should be corrected.

---

## Round 0.2 · accepted · Accept

The manuscript is interesting.

Reviewer 1 ·

Basic reporting

I already expressed my positive impression of the paper in my original review. All my comments were properly addressed, the dataset is publicly available, so I can't see any objections to publication.

Experimental design

Nothing to add compared to my original review

Validity of the findings

Nothing to add compared to my original review